# ROTATIONS ON LATENT HYPERSPHERES: A GEOMETRY-AWARE GUIDING FRAMEWORK FOR DIFFUSION MODELS

## ABSTRACT

Diffusion models have emerged as a powerful tool across diverse domains. However, their purely data-driven nature can produce samples that deviate from domain-governing constraints. We introduce a plug-and-play, Reinforcement Learning framework that operates in the latent space of pre-trained diffusion models to optimize initial noise samples. Our approach, motivated by the near-spherical geometry of high-dimensional Gaussian distributions, employs a novel rotation-matrix-based scheme for efficient latent space exploration. This steers the model toward more feature-preserving outputs, guided by task-specific rewards computed on the final samples. We evaluate our method on three diffusion models: one trained on solutions of the Darcy Flow PDE, one on a synthetic dataset with complex structural features, and a text-conditioned one. Across all three settings, our framework yields significant improvements in sample quality, achieving a $\sim 25\%$ relative reduction in PDE residual, up to a $\sim 44\%$ relative improvement on the synthetic dataset's feature-alignment metric, and up to a $\sim 80\%$ relative improvement on human preference, compared to the vanilla diffusion models. Finally, we show that rotation-matrix-based exploration significantly outperforms unconstrained exploration, validating our geometry-aware approach and establishing a more effective method for latent space control.

## 1 INTRODUCTION

Deep generative models, particularly diffusion (Song et al., 2021) and Latent Diffusion Models (LDM) (Rombach et al., 2022) have emerged as remarkably powerful tools for learning complex data distributions. Their success has been more prominent in image synthesis, where they have revolutionized computer vision and content creation. However, their applicability extends far beyond the visual arts, with successful deployments in scientific and engineering domains such as 3D modeling (Hu et al., 2024), audio synthesis (Prenger et al., 2019; Yamamoto et al., 2020; Kong et al., 2021), molecular generation (Gómez-Bombarelli et al., 2018; De Cao & Kipf, 2018; Zang & Wang, 2020; Sun et al., 2021; Xu et al., 2022), protein design (Repecka et al., 2021; Kozlova et al., 2023; Watson et al., 2023) , physics simulation (Jiang et al., 2021; Won et al., 2022; Holzschuh et al., 2023) and recently material design (Zeni et al., 2025).

Despite their impressive capabilities, a key limitation of these purely data-driven models is their tendency to produce samples that may not adhere to known, domain-governing constraints. This is particularly critical in scientific applications where outputs must satisfy physical laws, mathematical principles, or structural requirements. Furthermore, many applications benefit from steering the generative process towards a specific, desirable region of the output distribution for downstream tasks. Prominent examples include generating solutions to Partial Differential Equations (PDEs), where outputs must remain consistent with physical principles; designing 3D models with strict geometric tolerances; or creating novel proteins that adhere to fundamental biochemical constraints.

To address this, several techniques for guiding generative models have been proposed. These range from costly fine-tuning the model's weights or incorporating constraints directly into the training objective, to Latent Space Optimization (LSO). LSO has emerged as a flexible, post-hoc alternative that operates on the initial noise vectors of a pre-trained, frozen generative model, often guided

by reward signals to align outputs with specific objectives like human preferences. Nevertheless, within the current LSO landscape, we identify two significant gaps. First, most existing methods are computationally expensive, requiring iterative optimization and multiple full denoising passes to generate a single sample, which creates a substantial bottleneck at inference time. Second, current approaches often treat the latent space as a generic Euclidean space, overlooking the near-spherical geometry of high-dimensional Gaussian noise. We posit that failing to account for this inherent structure leads to inefficient exploration and can degrade sample quality.

In this work, we propose a novel framework to address these shortcomings. Our main contributions are:

- We introduce Rotational Latent Space Optimization (RLSO), a novel, geometry-aware, modular exploration technique for the latent space of frozen diffusion models. RLSO leverages rotation matrices to preserve the norm of latent vectors, respecting the inherent geometry of the Gaussian prior.

- We formulate the optimization of the initial noise as a Reinforcement Learning (RL) problem, allowing us to amortize the optimization cost. This enables the generation of constraint-aligned samples with only a single denoising pass at inference time, drastically improving efficiency.

- We provide a validation of our approach on three distinct use-cases from vastly different domains, demonstrating its versatility and effectiveness in improving sample quality.

Our experiments show that our geometry-aware RLSO framework significantly outperforms standard, unconstrained exploration. Furthermore, we demonstrate that the modularity of our approach can be leveraged to control the trade-off between generalization and computational efficiency, establishing a new and effective paradigm for latent space control in diffusion models.

## 2 RELATED WORK

**High-Dimensional Rotations**  The study of the theory and the application of Rotations in dimensions higher than three originates in the 18th century, but the literature remains limited and lacks a unified taxonomy. While some applications have successfully used the Rodrigues' formula (Rodrigues, 1840) or Cayley's transform (Cayley, 2009; 1846), for our work we use rotation matrices as a representation of rotations. Specifically, we follow Schoute (1892)'s theoretical generalization of Euler (1776)'s *Principal Rotation Theorem* to n dimensions:

*Any displacement of a rigid body about a fixed point in n dimensions can be achieved for $n$ even by $\frac{n}{2}$ simple rotations in mutually orthogonal planes about the fixed point and for $n$ odd by $\frac{n-1}{2}$ such rotations. Furthermore the rotations commute.*

In practice, we rely on Mortari (2001): by using the properties of the eigen-analysis of rotation matrices and an n-dimensional extension to the vector cross product (Mortari, 1997), the author provides a formula to construct simple rotations. This formulation allows us to uniquely identify a rotation matrix by defining a rotation angle and a principal plane of rotation.

**Latent Space of Diffusion Models**  We define the latent space of a Diffusion model as the Gaussian space $\mathcal{X}_0 \sim \mathcal{N}(\mathbf{0}, I)$ containing all possible initial noise samples $x_0 \in \mathbb{R}^{c \times N \times N}$, where $c$ is the number of channels and $N$ is the latent dimension. In higher dimensions ($N >> 1$), due to the *Concentration of Measure Phenomenon* (Wainwright, 2019) and the *Gaussian Annulus Theorem* (Blum et al., 2020), the expected length of Gaussian samples is concentrated around the square root of its dimensions $d$, i.e. in a thin shell of an n-sphere with radius $\sqrt{d}$. This means that a gaussian space behaves more akin to a hyper-spherical space than an euclidean one.

Though underexplored, this phenomenon has been described (Arvanitidis et al., 2018; Chen et al., 2018) and exploited, either by exploring the latent space with norm-regularization techniques or Spherical Linear Interpolation (Videau et al., 2023; Samuel et al., 2023; Bodin et al., 2024; Sacchetto et al., 2024). The works of Park et al. (2023) and Jin et al. (2025) advance in this direction and use geodesic shooting for latent space exploration and Rodrigues' formula-based rotation for guidance, respectively.

We note that, while our work focuses on diffusion models specifically, this interpretation is valid for any kind of generative model that possesses a Gaussian latent space.

**Latent Space Optimization**  Latent space Optimization, or Noise Optimization, is a rapidly growing field that encompasses all algorithms and techniques which aim to optimize the input noise (or some intermediary latent space) of frozen, pre-trained generative models. A common approach is to optimize latent samples with classic optimization techniques, either by selecting the best candidate out of a population (Karthik et al., 2023) or by backpropagating the gradients through the full denoising process (Samuel et al., 2023; Wallace et al., 2023; Samuel et al., 2024; Karunratanakul et al., 2024). Alternatively, Eyring et al. (2024) use a one-step diffusion model. These do achieve significant improvements, but have two significant drawbacks: backpropagating the gradients through the denoising process may be costly and importantly they add significant computational overhead at inference time, because they need to run the denoising process multiple times (Wu et al., 2023).

On the other hand, this issue can also be mitigated by training an auxiliary model to optimize the latent samples, so that only one pass through the auxiliary model and the denoising process is required at inference time. Lu et al. (2023) use an auxiliary model to predict the values of an energy function to guide the sampling process, Ahn et al. (2024) train a model in the latent space to mimic Classifier-Free Guidance, while Venkatraman et al. (2025) train a model to substitute the sampler by learning the reverse denoising process of high-reward samples. Most similarly to our work, Eyring et al. (2025) recently proposed to train a LoRA network to predict latent samples that denoise into high-reward samples.

While these papers presents similarities to our work, namely training auxiliary models to generate optimized latent samples for pre-trained diffusion models, our approach introduces several key differences. Typically, the techniques employed to ensure that optimized latent samples remain Gaussian and within the *shell* range from adding a penalty term to the cost function (Eyring et al., 2024; 2025; Venkatraman et al., 2025), adding a small Gaussian perturbation to each update step (Karunratanakul et al., 2024), projecting back to the shell and adding small Gaussian perturbations (Wallace et al., 2023), or using small, regularized gradient steps (Karunratanakul et al., 2024). Instead of relying on penalizing terms with high overhead or projecting back to the shell, which limits exploration and doesn't ensure Gaussianity on its own, our rotation-matrix-based exploration strategy offers a principled and geometry-aware method for generating optimized latent samples that automatically remain both in-distribution and semantically linked to the original latent sample.

Moreover, this exploration strategy allows for modular control over the direction and angle. In contrast to Ahn et al. (2024) and Eyring et al. (2025), our use of Policy Gradient training enables compatibility with arbitrary reward functions, even when their gradients are intractable. Compared to Venkatraman et al. (2025), who train a large U-Net diffusion model, our approach is far more lightweight, requiring an order of magnitude fewer parameters. Finally, we validate our method beyond text-to-image models and human preference alignment, demonstrating its effectiveness across diverse domains.

Approaches whose noise optimization is an inherent part of training the main model naturally do not suffer from this issue, like Hu et al. (2025) who train the encoder of their encoder-decoder structure as an RL-policy in the latent space or Wagenmaker et al. (2025) who train an RL policy for robot control tasks that outputs actions in the latent space. Finally, Zhang et al. (2025) expand on the work of Lu et al. (2023) by integrating the energy-function guidance in the training of the main model.

## 3  METHODOLOGY

Our Approach employs a Reinforcement Learning Agent to navigate the latent space of a frozen, pre-trained Latent Diffusion Model. The agent is effectively trained to apply a rotational tranformation to the initial Gaussian noise sample, before the LDM decodes it into a final sample (see Appendix A.2). To serve as engine and benchmark to our experiments we select two model architectures: an unconditional Latent Diffusion Model and a text-conditioned Latent Diffusion Model.

## 3.1 LATENT DIFFUSION MODELS

### 3.1.1 UNCONDITIONAL LATENT DIFFUSION MODEL

We train an unconditional Latent Diffusion model (Rombach et al., 2022) from scratch on two datasets from different domains: solutions of the Darcy Flow Partial Differential Equations (PDE) and an ad-hoc synthetic image dataset. Both provide computable metrics that measure how much a sample violates the dataset validity constraints, i.e. a *residual error*. A residual error of zero corresponds to a valid sample.

**Darcy Flow** Here we use the dataset created by Bastek et al. (2025), based partially on the work of Jacobsen et al. (2025): it is a dataset of 10000 solutions of the steady-state 2D Darcy-flow PDEs, which describe fluid movement through a porous medium. Each of the samples is generated by sampling the permeability field $K(\xi)$ from a Gaussian random field on a $64 \times 64$ grid and solving for the pressure distribution $p(\xi)$ with a finite-differences, least-squares linear solver. This results in samples $(K, p) \in \mathbb{R}^{2 \times 64 \times 64}$. The per-grid-cell residual error is calculated based on the physical law of mass conservation as follows:

$$R(K, p) = \nabla(K\nabla p) + f. \tag{1}$$

where $K$ is the permeability field, $p$ is the pressure field, and $f$ is the source function. A scalar residual error for one sample $x_0$ is then obtained as the mean absolute residual error:

$$\epsilon(x_0) = \frac{1}{n^2} \sum_{i=1}^{n} \sum_{j=1}^{n} |R_{ij}(K, p)| \tag{2}$$

where $n = 64$ and $R_{ij}(K, p)$ is the residual error at grid cell $(i, j)$. We refer to Bastek et al. (2025) for further details.

**Voronoi** Deshpande et al. (2024) introduce several synthetic datasets to benchmark image-synthesis generative models. These datasets, which the authors call Stochastic Context Models (SCM), contain images with different features, constraint, and rules that can be recovered after generation. They also provide scripts to compute a variety of quality metrics based on these features. For our experiments, we select the Voronoi SCM and simplify it slightly, using 64x64 instead of 256x256 images and restricting it to the class of images containing 16 regions. Hence creating a dataset of 10000 grayscale images. Out of a selection of the quality metrics introduced by the authors we define our own residual error:

$$\epsilon(x_0) = \frac{\mu_1}{A} + \frac{\sigma_1}{2B} + \frac{\mu_2}{\Gamma} + \frac{\sigma_2}{2\Delta} + \tau + \rho + 1.5\eta, \tag{3}$$

where $\mu_1$ and $\sigma_1$ are measures of the straightness of region edges, $\mu_2$ and $\sigma_2$ are measures of the intra-region grayscale variance, $\tau$ and $\rho$ are Kendall's and Spearman's rank correlation coefficients between the region's and the target grayscale values, and $\eta$ is the error in region count. Furthermore, $A$, $B$, $\Gamma$, and $\Delta$ are normalizing constants. Specifically, $A = 0.0962016$ and $B = 0.116852$ are the average $\mu_1$ and $\sigma_1$ of the training dataset, respectively. $\Gamma = 50$ and $\Delta = 20$ are set such that $\sim 95\%$ of the non-zero values of the training dataset fall in the $[0, 1]$ range.

The implementation of the latent diffusion model was adapted from (von Platen et al., 2022) and modified to include a Variational Autoencoder (VAE) with KL loss (Kingma & Welling, 2013). We employ DDIM (Song et al., 2020) as our sampler. Architecture and training details are listed in the Appendix A.

### 3.1.2 TEXT-CONDITIONED LATENT DIFFUSION MODEL

For the text-conditioned LDM, we use Stable Diffusion 1.5 (Rombach et al., 2022), integrated with the 2-Step Hyper-SD LoRA (Ren et al., 2024) and the DDIM sampler (Song et al., 2020). As a metric for sample quality, we employ Image Reward (Xu et al., 2023), a pre-trained text-to-image human preference reward model that, given a generated image and its corresponding prompt, outputs a human preference score. Differently to the experiments on the unconditional LDM, the human preference score is not a residual since a higher score corresponds to a higher sample quality and it has no theoretical upper bound. Therefore, we define the reward as the output of the human preference reward model $G(x_0)$, shifted so that the majority of rewards are negative:

$$r(x_0) = G(x_0) - 2 \tag{4}$$

## 3.2 ROTATION MATRIX

**Theoretical Framework**  We define a parameter description of n-dimensional rotations that both encompasses all mathematic properties of rotation and offers modularity for managing the tradeoff between generalization capabilities and computational efficiency. To this end, we adopt the theoretical framework described in chapter 2, which posits that a general rotation in even dimensions can be described by $\frac{n}{2}$ mutually orthogonal planes (the rotation planes) and corresponding $\frac{n}{2}$ angles. Based on this framework, Lounesto (2001) identifies three special classes of rotations. These are:

- single rotations: only one plane of rotation with angle $\theta \neq 0$.
- double rotations: two planes of rotation with angles $\alpha \neq \theta \neq 0$.
- isoclinic rotation: two planes of rotation with angles $\alpha = \theta \neq 0$.

We extend this classification system to higher dimensions and introduce the n-fold rotation. In n dimensions, there exists up to n-fold rotations. They can be isoclinic (if n is even) or as pseudo-isoclinic (if n is odd) Richard et al. (2010). In his paper, (Mortari, 2001, eq. 18) provides a formula for a rotation matrix that describes a single rotation as a function of the rotation angle and the vectors spanning the plane of rotation:

$$R(P, \Phi) = I_n + (\cos \Phi - 1)PP^T + P \begin{bmatrix} 0 & -1 \\ 1 & 0 \end{bmatrix} P^T \sin \Phi \tag{5}$$

Where $\Phi$ is the angle of rotation and $P = \begin{bmatrix} \mathbf{p_1} & \mathbf{p_2} \end{bmatrix} \in \mathbb{R}^{n \times 2}$ is a matrix whose columns form an orthogonal basis for the plane of rotation. Using the fact that a general rotation can be expressed as a product of $\frac{n}{2}$ simple rotations, we extend equation 5 for rotation matrices of general rotations.

$$R_{\text{gen}} = \prod_{i=1}^{\frac{n}{2}} R(P_i, \Phi_i) \tag{6}$$

Where $\begin{bmatrix} P_1 & P_2 & \dots & P_{\frac{n}{2}} \end{bmatrix} \in \mathbb{R}^{nxn}$ is an orthonormal basis of $\mathbb{R}^n$ and $\begin{bmatrix} \Phi_1 & \Phi_2 & \dots & \Phi_{\frac{n}{2}} \end{bmatrix} \in \mathbb{R}^n$ are the rotation angles. We note that equation 6 simplifies to equation 5 for only one $\Phi_i \neq 0$.

**Vector Rotation**  Let $\mathbf{v} \in S^{n-1} \subset \mathbb{R}^n$ a point on the surface of the sphere $S^{n-1}$ and $\hat{\mathbf{v}} = \frac{\mathbf{v}}{\|\mathbf{v}\|}$ its corresponding unit vector. let $\mathbf{t} \in T_{\mathbf{v}}(S^{n-1})$ be a non-zero unit vector in the tangent space at $\mathbf{v}$. To rotate $\mathbf{v}$ towards $\mathbf{t}$ along a geodesic by an angle $\Phi$ we can use equation 5 with $P := \begin{bmatrix} \hat{\mathbf{v}} & \mathbf{t} \end{bmatrix}$:

$$\mathbf{v_{rot}} = R(\begin{bmatrix} \hat{\mathbf{v}} & \mathbf{t} \end{bmatrix}, \Phi)\mathbf{v} \tag{7}$$

where $\mathbf{v_{rot}}$ is the rotated vector.

Now let $\mathbf{w}$ be a second vector in the tangent space at $\mathbf{v}$, orthogonal to $\mathbf{t}$. To perform a double rotation on $\mathbf{v}$ in the directions of $\mathbf{t}$ and $\mathbf{w}$, we can use 6:

$$\mathbf{v_{rot}} = R(\begin{bmatrix} \hat{\mathbf{v}} & \mathbf{t} \end{bmatrix}, \Phi_1)R(\begin{bmatrix} \hat{\mathbf{v}} & \mathbf{w} \end{bmatrix}, \Phi_2)\mathbf{v} \tag{8}$$

Similarly, we can construct rotations with any number of planes and angles, up to a general rotation. For the purposes of this work, we restrict the rotations to paths along geodesics (or combinations thereof). This has two key advantages:

1. We can describe the rotation of a vector with $k$ $n$-dimensional direction vectors and $k$ angles, i.e. $k(n-1)$ parameters. For a simple rotation, this is equivalent to other retraction methods.

2. Because random vectors in high dimensions are always almost orthogonal (Diaconis & Freedman, 1984), this constraint helps prevent the optimization from exploring rotations that have little to no effect on the vector's position.

This formulation provides significant modularity. Unlike common exploration techniques such as back-projection or the exponential map, our approach decouples the angle and direction of rotation. This allows for fine-grained control over the scope and nature of the directional exploration. Furthermore, by enabling precise manipulation of the number and angles of rotation planes, our method facilitates the construction of more complex rotational transformations than previously possible.

**A note on meaningful change**  Because residual errors are computed on the entire sample, we are primarily interested in transformations that affect a large portion of a latent vector's dimensions. However, simple rotations do not consistently achieve this. For example, individual Givens rotations (Givens, 1958), i.e. rotations confined to hyperplanes spanned by coordinate axes, only modify the corresponding coordinate pair. They therefore induce negligible change in high-dimensional settings, causing the LDM to denoise the transformed latent sample into one nearly identical to the original, except for a small localized change. At the opposite extreme, one can show that rotations defined by planes spanned by vectors maximally distant from the coordinate axes impact the greatest number of dimensions of an $n$-dimensional vector. To restrict exploration to rotations that meaningfully change latent vectors, we construct a set of $k$ fixed directions in tangent space. For $k \leq n$, we select the columns of a Hadamard matrix, as they are simultaneously maximally distant from all coordinate hyperplanes and mutually orthogonal Tadej & Życzkowski (2006). For $k > n$, we use a gradient-based energy minimization algorithm that iteratively refines randomly initialized points. The optimization objective combines a repulsive force to maximize angular separation and a penalty to repels vectors from coordinate hyperplanes. For our setting, we project these column vectors onto the tangent space of the sphere at $v$ and normalize them.

### 3.3 Reinforcement Learning

The RL Problem is defined as a Markov decision process (MDP) characterized by the tuple $(S, A, P, r, \gamma)$; where $S$ is the state space, $A$ is the action space, $P(s'|s, a)$ is the system transition probability, $r(s, a)$ is the reward, and $\gamma \in (0, 1)$ is the discount factor. The goal of the RL algorithm is to find an optimal policy $\pi^*(a|s)$ that maximizes the expected cumulative discounted reward:

$$\pi^* = \arg\max_{\pi} \mathbb{E}_{\pi} \left[ \sum_{t=0}^{\infty} \gamma^t r(s_t, a_t) \right] \tag{9}$$

**Observation Space**  the observation space is equal for all experiments. At each time-step $t$ the Agent receives observation $s_t = \text{vec}(x_0) \in \mathbb{R}^{c \times d^2} \sim \mathcal{N}(0, I)$, which is a latent sample with its spatial dimensions flattened, where $d$ is the latent dimension and $c$ is the number of channels. Specifically:

- Unconditional LDM - Darcy Flow: $c = 2$, $d = 16 \Rightarrow s_t \in \mathbb{R}^{512}$.

- Unconditional LDM - Voronoi: $c = 1$, $d = 16 \Rightarrow s_t \in \mathbb{R}^{1024}$.

- Text-conditioned LDM: $c = 4$, $d = 64 \Rightarrow s_t \in \mathbb{R}^{16384}$.

**Reward Function**  The reward function is dependent on the architecture. For the unconditional LDM, we define the reward function as the negative residual of the respective dataset:

$$r(s, a) = -\epsilon(x_0) \tag{10}$$

where $x_0 = s$ and $\epsilon(x_0)$ is computed according to Equation 2 for the Darcy Flow dataset and according to Equation 3 for the Voronoi dataset. For the text-conditioned LDM, we set the reward function equal to the reward defined in equation 4.

An episode terminates either upon reaching a predefined reward threshold or after 15 time-steps. To set the reward thresholds, we generated 10000 samples using the LDM and selected the value exceeded by the top $5\%$ of samples. This resulted in values of $\tau = -0.4$ for the Darcy Flow dataset, $\tau = -1.5$ for the Voronoi dataset, and $\tau = -0.2$ for the text-conditioned LDM.

The action space varies depending on the experiment and is discussed in section 4. To train the agent, we employ Proximal Policy Optimization (PPO) (Schulman et al., 2017), a state-of-the art on-policy policy optimization algorithm. While the goal of the policy is not necessarily to find trajectories to optimal samples, but rather to identify them in one or a few steps, this setup effectively mimics a multi-armed bandit problem. Nonetheless, we choose PPO due to its superior ability to handle high-dimensional and partially continuous observation and action spaces, which are required in our experiments.

## 4 EXPERIMENTS

The experimental setup aims to validate three main hypotheses:

**Hypothesis 1** *Reinforcement learning is a viable paradigm for performing amortized, gradient-free optimization in the latent space of frozen diffusion models.*

**Hypothesis 2** *Exploration techniques that account for the inherent spherical geometry of Gaussian latent spaces significantly outperform naive approaches.*

**Hypothesis 3** *An exploration technique that enables control over the trade-off between generalization capabilities and computational efficiency (i.e., the number of parameters) offers significant advantages.*

To this end, we conducted 5 experiments: three analogous ones on both the Voronoi and the Darcy Flow datasets, one only on the Darcy flow dataset, and one on the text-conditioned LDM. Generally, the only variables between experiments are the action space, the observation space, and the reward function. We refer to appendix A for all other architecture and implementation details. For all experiments, we applied the same transformation to all channels. Since the channels encode spatial information, this approach ensures that the transformations do not disrupt the spatial relationships learned by the VAE.

### 4.1 EXPERIMENT 1 - UNCONSTRAINED EXPLORATION

To serve as our benchmark we define the action space of the RL-Agent such that it can move freely in the latent space. Therefore, the actions lie in the space $(\alpha, \mathbf{u})$, where $\alpha \in \mathbb{R}$ is a scalar, $\mathbf{u} \in \mathbb{R}^{n^2}$ is a unit vector, and $w = \alpha\mathbf{u} \in \mathbb{R}^{n^2}$ is perturbation vector that is summed to all channels. The next state is computed as $\mathbf{s}_{t+1} = \mathbf{s}_t + \mathbf{1}_c \otimes \mathbf{w}$, where $\mathbf{1}_c \in \mathbb{R}^c$ is a vector of ones corresponding to the number of channels, and $\otimes$ denotes the outer product, so that $w$ is added to each channel of $\mathbf{s}_t$.

### 4.2 EXPERIMENT 2 - ROTATION MATRIX I

Instead of moving freely in the latent space, here the RL-Agent is constrained to the n-sphere. It moves by choosing a direction and an angle of rotation. Therefore, the actions lie in the space $(\hat{t}, \Phi)$, where $\hat{t} \in T_s(S^{d^2-1}) \subset \mathbb{R}^{d^2}$ is a unit vector in the tangent space of the sphere at $s$ and $\Phi \in [0, \pi]$ is a rotation angle. The next state is then computed according to equation 7 on all channels:

$$\mathbf{s}_{t+1} = \begin{bmatrix} R\left(\begin{bmatrix} \hat{\mathbf{s}}_{\mathbf{t,1}} & \hat{\mathbf{t}} \end{bmatrix}, \Phi\right) \mathbf{s}_{t,1} & \cdots & R\left(\begin{bmatrix} \hat{\mathbf{s}}_{\mathbf{t,1}} & \hat{\mathbf{t}} \end{bmatrix}, \Phi\right) \mathbf{s}_{t,n} \end{bmatrix}$$

where $\mathbf{s}_{\mathbf{t,i}}$ denotes channel $i$ of $s_t$, , and $R$ is the rotation matrix as defined in Equation 5.

### 4.3 EXPERIMENT 3 - ROTATION MATRIX II

Here, we investigate the effects of discretizing the action space. Instead of choosing a vector of continuous values, the RL-agent moves by selecting one of a predetermined, fixed number of direction vectors and an angle step. The actions lie in the space $(j, \Phi)$, where $j \in 1, \ldots, N_d$ and $\Phi = \frac{p\pi}{79}$, for $p \in 0, \ldots, 79$. The next state is then computed according to equation 7:

$$\mathbf{s}_{t+1} = \begin{bmatrix} R\left(\begin{bmatrix} \hat{\mathbf{s}}_{\mathbf{t,1}} & \hat{\mathbf{h}_{\mathbf{j}}} \end{bmatrix}, \Phi\right) \mathbf{s}_{t,1} & \cdots & R\left(\begin{bmatrix} \hat{\mathbf{s}}_{\mathbf{t,1}} & \hat{\mathbf{h}_{\mathbf{j}}} \end{bmatrix}, \Phi\right) \mathbf{s}_{t,n} \end{bmatrix}$$

where $\hat{\mathbf{h}_{\mathbf{j}}}$ is th $j$th column of the direction matrix $H \in \mathbb{R}^{d^2 \times N_d}$, $\Phi$ the discretized angle, $\mathbf{s}_{\mathbf{t,i}}$ denotes channel $i$ of $s_t$, and $R$ is the rotation matrix as defined in Equation 5.

### 4.4 EXPERIMENT 4 - DOUBLE ROTATION

In experiments two and three, we have computed the rotation only from one channel and have applied it to both. In this experiment, to analyzes the effect of making the action dependent on the entire observation space, we compute the rotation from all channels and keep the action space discrete. Therefore, the actions lie in the space $(j, k, \Phi)$, where $j, k \in 1, \ldots, d^2$ and

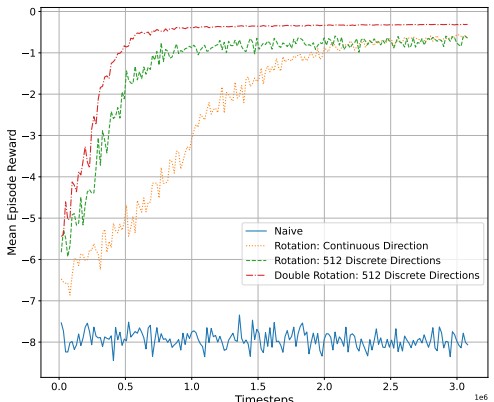 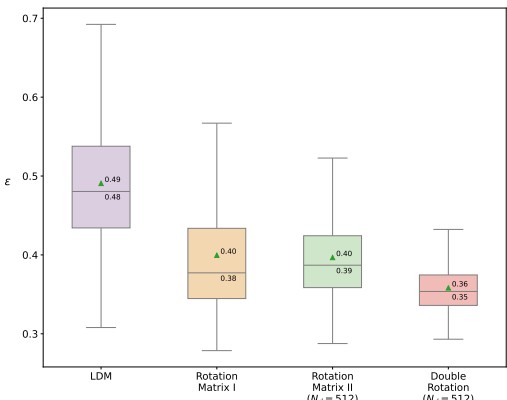

Figure 1: Darcy Flow: RL training (left) and residual error comparison (right), evaluated on 10000 samples

$\Phi = \frac{p\pi}{79}$, for $p \in 0, \ldots, 79$. We essentially perform a isoclininc, double rotation with two planes of rotations: $\mathrm{Span}(\mathbf{s_{t_1}}, \hat{\mathbf{h}_j})$ and $\mathrm{Span}(\mathbf{s_{t_2}}, \hat{\mathbf{h}_k})$. The next state is then computed according to equation 8:

$$\mathbf{s}_{t+1} = \begin{bmatrix} R^{(2)}\mathbf{s}_{t,1} & R^{(2)}\mathbf{s}_{t,2} \end{bmatrix}$$

where:

$$R^{(2)} = R\left(\begin{bmatrix} \hat{\mathbf{s}}_{\mathbf{t,1}} & \hat{\mathbf{h}_j} \end{bmatrix}, \Phi\right) R\left(\begin{bmatrix} \hat{\mathbf{s}_{t,2}} & \hat{\mathbf{h}_k} \end{bmatrix}, \Phi\right)$$

with $\hat{\mathbf{h}_j}$ and $\hat{\mathbf{h}_k}$ denoting columns $j, k$ of the direction matrix $H \in \mathbb{R}^{d^2 \times N_d}$, $\Phi$ the discretized angle, $\mathbf{s_{t,i}}$ the channel $i$ of $s_t$, and $R$ the rotation matrix as defined in Equation 6.

### 4.5 EXPERIMENT 5 - TEXT-CONDITIONED LDM

Here, the latent space is significantly larger (16384 dimensions). To keep the action space exploding in dimensionality, we keep the action space discrete with a limited amount of fixed directions. Furthermore, similarly to experiment four, we compute the rotation from all channels. Therefore the actions lie in the space $(j, k, l, m, \Phi_1, \Phi_2, \Phi_3, \Phi_4)$, where $j, k, l, m \in 1, \ldots, 512$ and $\Phi_i = \frac{p\pi}{79}$, for $p \in 0, \ldots, 79$. I.e., we perform a 4-fold rotation with four planes of rotations:

$$\mathbf{s}_{t+1} = \begin{bmatrix} R^{(4)}\mathbf{s}_{t,1} & R^{(4)}\mathbf{s}_{t,2} & R^{(4)}\mathbf{s}_{t,3} & R^{(4)}\mathbf{s}_{t,4} \end{bmatrix}$$

where:

$$R^{(4)} = R\left(\begin{bmatrix} \hat{\mathbf{s}}_{\mathbf{t,1}} & \hat{\mathbf{h}_j} \end{bmatrix}, \Phi\right) R\left(\begin{bmatrix} \hat{\mathbf{s}_{t,2}} & \hat{\mathbf{h}_k} \end{bmatrix}, \Phi\right) R\left(\begin{bmatrix} \hat{\mathbf{s}}_{\mathbf{t,3}} & \hat{\mathbf{h}_l} \end{bmatrix}, \Phi\right) R\left(\begin{bmatrix} \hat{\mathbf{s}_{t,4}} & \hat{\mathbf{h}_m} \end{bmatrix}, \Phi\right)$$

with $\hat{\mathbf{h}_j}$, $\hat{\mathbf{h}_k}$, $\hat{\mathbf{h}_l}$, and $\hat{\mathbf{h}_m}$ denoting columns $j, k, l, m$ of the direction matrix $H \in \mathbb{R}^{d^2 \times N_d}$; $\Phi_1$, $\Phi_2$, $\Phi_3$, $\Phi_4$ the discretized angles of the respective planes of rotation; $\mathbf{s_{t,i}}$ the channel $i$ of $s_t$; and $R$ the rotation matrix as defined in Equation 6. We train with the fixed prompt: *"a photo of an astronaut riding a horse on mars"*.

## 5 RESULTS AND DISCUSSION

### 5.1 UNCONDITIONAL LDM

**Darcy Flow**  Figure 1 presents the results of experiments 1, 2, 3, and 4 conducted on the Darcy Flow dataset. The left subfigure highlights two key findings: Firstly, our geometry-aware exploration technique consistently outperforms unconstrained exploration, thereby validating hypothesis 2. Second, restricting the search directions to a predefined set led to faster and, in some cases, higher convergence. Furthermore, by leveraging the chosen theoretical framework of n-dimensional rotations, and using the underexplored concepts of double and isoclinic rotations, we were able to further enhance exploration performance, validating hypothesis 3. The right subfigure provides evidence for

hypothesis 1: our framework achieved a $\sim 25\%$ relative reduction in PDE residual compared to the vanilla LDM and obtained an absolute PDE residual comparable to state-of-the-art models, outperforming CoCoGen (Jacobsen et al., 2025) and PG-diffusion (Shu et al., 2023) but trailing PIDM (Bastek et al., 2025), as reported by Bastek et al. (2025).

**Voronoi**   Figure 2 presents the results of experiments 1,2, and 3 conducted on the Voronoi dataset. The findings closely parallel those from the Darcy Flow experiments: geometry-aware exploration consistently outperforms unconstrained exploration, resulting in an approximate $\sim 10\%$ relative reduction in residual error for experiment 2 and an approximate $\sim 44\%$ relative reduction for experiment 3.

**Diversity**   We have conducted an extensive analysis of quality and diversity for both datasets (see appendix B). The results show that diversity depends on the size of the action space and of the latent space. In all our experiments, a continuous action space never resulted in a loss in diversity. In the Darcy Flow experiments, a discrete action space with $N_d = 256$ fixed directions lead to a slight reduction in diversity, which didn't appear with $N_d = 512$ directions. For the Voronoi dataset, which has the smallest latent space, discretizing the action space always led to loss in diversity, albeit on different levels depending on the type of discretization.

## 5.2   TEXT-CONDITIONED LDM

Figure 3 presents the results for experiment 5 conducted on the text-conditioned LDM. Similar to the unconditional LDM experiments, RLSO significantly improves the sample quality based on the target metric: in this case, the human preference score from the Image Reward model. We observed up to an $\sim 80\%$ relative improvement when using the same fixed prompt as in training. Furthermore, we report a $\sim 12\%$ relative improvement with a different fixed prompt and a $\sim 3.7\%$ relative improvement with random prompts from a small prompt dataset (see appendix B.3), indicating that the model also generalizes to unseen conditionings. This experiment demonstrates the importance of the modularity of RLSO. Despite the significant increase in the dimensionality of the latent space (Stable Diffusion 1.5 has a latent space $\mathcal{X} \in \mathbb{R}^{64 \times 64 \times 4}$ ), the number of training step did not significantly increase. This is due to two main reasons: first, training time scales primarily with the observation space, which we can control independently of dimensionality by adjusting the number of directions; second, complexity scales only minimally with the number of channels due to the use of n-fold rotations.

In their paper, Venkatraman et al. (2025) introduce Outsourced Diffusion and include a comparison to previous works that use RL to train or fine-tune diffusion models. These are DDPO (Black et al., 2024), DPOK (Fan et al., 2023), and RTB (Venkatraman et al., 2024). The authors report the average ImageReward score and the average diversity score (measured as the mean cosine distance between CLIP embeddings) for each model across four fixed prompts. Although the comparison is limited,

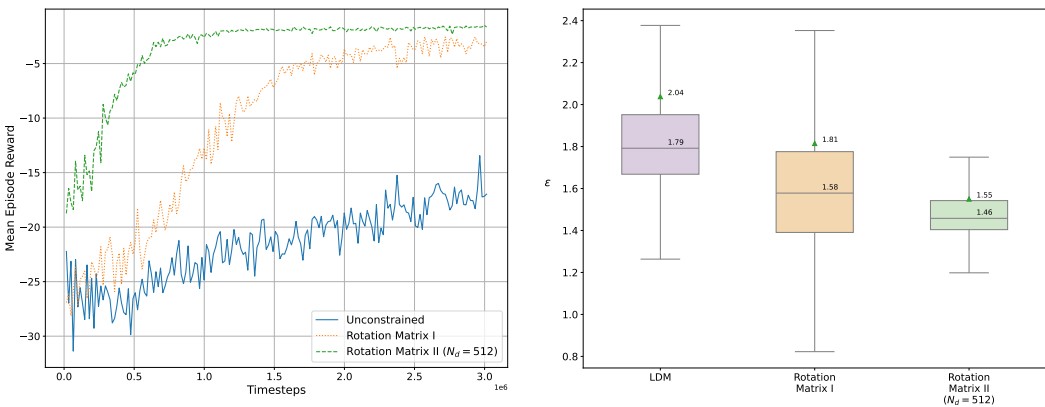

Figure 2: Voronoi: RL training (left) and residual error comparison (right), evaluated on 10000 samples

| Model | Backbone | Reward | CLIP Diversity |
|---|---|---|---|
| Prior | SD1.5 | $-0.17$ | 0.18 |
| DDPO | SD1.5 | 1.37 | 0.09 |
| DPOK | SD1.5 | 1.23 | 0.13 |
| RTB | SD1.5 | **1.4** | 0.11 |
| Outsourced Diff. | SD1.5 | 1.26 | 0.14 |
| Prior (ours) | SD1.5+Hyper-SD | 0.08 | 0.15 |
| RLSO (ours) | SD1.5+Hyper-SD | 1.2 | **0.17** |

Table 1: Alignment performance of RLSO on an accelerated backbone (Hyper-SD) compared to standard benchmarks on vanilla Stable Diffusion 1.5., extended from table 13 of Venkatraman et al. (2025)

because we used Stable Diffusion 1.5 with the Hyper-SD LoRA as the backbone of RLSO, while the baselines used vanilla Stable Diffusion 1.5, the absolute scores and the net gain over the prior shown in Table 1 demonstrate effective alignment.

## 6 CONCLUSION

In this work, we introduced Rotational Latent Space Optimization (RLSO) with Reinforcement Learning (RL), a novel framework that leverages reinforcement learning for efficient, geometry-aware guidance of pre-trained diffusion models. Our experiments, conducted across three diverse scientific domains, demonstrate that respecting the spherical geometry of the latent space via rotational exploration produces significantly higher-quality samples compared to unconstrained optimization. Moreover, by amortizing the optimization process with an RL agent, our method generates constraint-aligned samples in a single denoising pass, eliminating the substantial computational overhead of iterative LSO techniques. We have shown that this approach is a viable and effective paradigm for improving sample quality wherever a guiding reward signal is available, establishing a new path for efficient latent space control Furthermore, we have shown the modularity of our rotational approach facilitates optimization in lower-dimensional subspaces, ensuring the method scales effectively even as latent space dimensionality and model complexity increases increases.

**Limitations** RLSO is inherently shaped by the expressive capacity of the underlying diffusion model. It is particularly effective at identifying optimal latent codes within the learned data manifold, but it cannot introduce features entirely outside the model's training distribution. While our current implementation utilizes PPO for its stability in high-dimensional spaces, future work could explore more sample-efficient RL paradigms to further improve scalability.

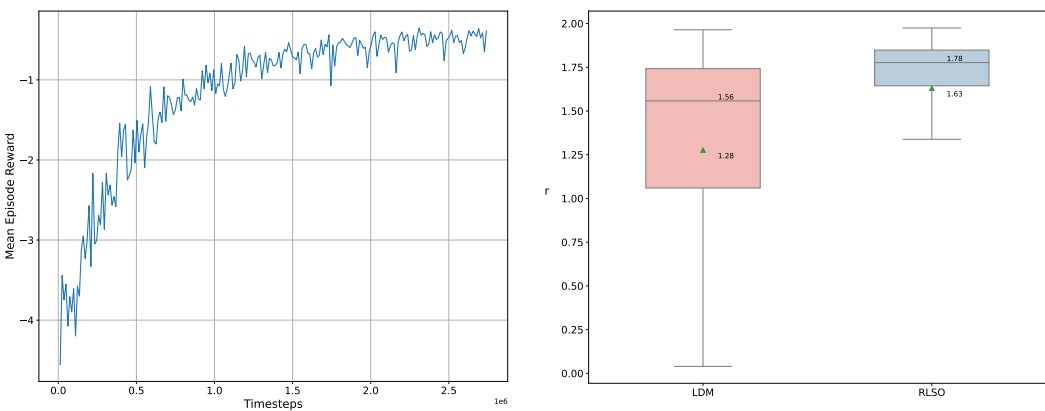

Figure 3: text-conditioned LDM: RL training (left) and Image Reward score (right), evaluated on 10000 samples conditioned with the fixed prompt: *"a photo of an astronaut riding a horse on mars"*.

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

# A  IMPLEMENTATION DETAILS

## A.1  HYPERPARAMETERS

| Hyperparameter | Value |
| --- | --- |
| In-, output channels (Darcy Flow) | $2, 2$ |
| In-, output channels (Voronoi) | $1, 1$ |
| In-, output channels (Stable Diffusion) | $4, 4$ |
| ResNet blocks per down-/up-sampling | 2 |
| ResNet block normalization | Group Normalization |
| ResNet block activation function | SiLU |
| Attention block normalization | LayerNorm |
| Feature map resolutions | $[64, 64, 128, 256]$ |
| Attention head dimension | 32 |

Table 2: Diffusion model architecture

| Hyperparameter | Value |
| --- | --- |
| Latent channels (Darcy Flow) | 2 |
| Latent channels (Voronoi) | 1 |
| Latent channels (Stable Diffusion) | 4 |
| Latent dimension (unconditional LDM) | $16 \times 16$ |
| Latent dimension (Stable Diffusion) | $64 \times 64$ |
| Down block type, number | DownEncoderBlock2D, 3 |
| Up block number, number | UPDecoderBlock2D, 3 |
| Block output channels | $[64, 64, 64]$ |

Table 3: VAE architecture

| Hyperparameter | Value |
| --- | --- |
| Actor Hidden layers | $[2048, 2048]$ |
| Critic Hidden Layers | $[2048, 2048]$ |
| Dropout Rate | 0.3 |
| Learning Rate | Linear Schedule $[1 \cdot 10^{-6}, 1 \cdot 10^{-7}]$ |
| GAE Lambda | 0.9 |

Table 4: PPO hyperparameters

## A.2 ALGORITHM DETAILS

---

**Algorithm 1** Rotational Latent Space Optimization (RLSO)

---

 1: **Input:** Pre-trained Latent Diffusion Model $D(\cdot)$, reward function $r(\cdot)$
 2: **Initialize:** Policy network $\pi_\theta(a|s)$ with parameters $\theta$.

---

    *Training*

---

 3: **procedure** TRAINING
 4:    **for** each training iteration **do**
 5:        Sample initial latent vector $s_0 \sim \mathcal{N}(0, I)$.
 6:        **for** $t = 0, \ldots, T - 1$ **do**               ▷ Loop over steps in an episode
 7:            Sample action $a_t \sim \pi_\theta(\cdot|s_t)$.
 8:            Construct rotation matrix $R(a_t)$ and compute the new latent vector: $s_{t+1} = Rs_t$.
 9:            Generate sample by decoding the latent vector: $y_{t+1} = D(s_{t+1})$.
10:            Compute reward: $r_{t+1}(y_{t+1})$.
11:            **if** episode terminates **then**
12:                **break**
13:        Update policy $\theta$ using PPO with the collected trajectory.

---

    *Inference*

---

14: **procedure** INFERENCE
15:    **Input:** Trained policy $\pi_{\theta^*}$, initial latent vector $s_{init} \sim \mathcal{N}(0, I)$.

16:    Sample initial latent vector $s_0 \sim \mathcal{N}(0, I)$.
17:    Compute optimized latent vector $s_{opt} = \pi_\theta(\cdot|s_0)$.
18:    Generate the final, optimized sample: $y_{opt} = D(s_{opt})$.
19:    **return** $y_{opt}$.

---

## A.3 TRAINING

| Experiment | GPU | Training steps | Hours |
|---|---|---|---|
| exp 2 - Darcy Flow | 1x Nvidia GeForce RTX 4090 | $2 \times 10^6$ | ∼14 |
| exp 2 - Voronoi | 1x Nvidia GeForce RTX 4090 | $1.5 \times 10^6$ | ∼11 |
| exp 3 - Darcy Flow ($N_D = 512$) | 1x Nvidia GeForce RTX 4090 | $1 \times 10^6$ | ∼7.8 |
| exp 3 - Voronoi ($N_D = 512$) | 1x Nvidia GeForce RTX 4090 | $0.8 \times 10^6$ | ∼6.7 |
| exp 4 - Darcy Flow ($N_D = 512$) | 1x Nvidia GeForce RTX 4090 | $0.6 \times 10^6$ | ∼5.2 |
| exp 5 - Stable Diffusion | 1x Nvidia GeForce RTX 5090 | $1.5 \times 10^6$ | ∼54 |

Table 5: Hardware details and training time (one training step comprises one full denoising pass of the frozen LDM and one update of the RL Policy's network weights).

## B TRADEOFF BETWEEN QUALITY AND DIVERSITY

To investigate the tradeoff between quality and diversity, we run experiments that analyze the effect of several variables on the diversity and on the quality of the generated samples. Specifically, the number of directions $N_d$ (continuous, 256, or 512); the angle range ($(0, \pi)$, $[0, \pi]$ or $[0, \frac{\pi}{2}]$); the number of discretization bins $B_\Phi$ (80 or 160); and the type of rotation (single or double).

### B.1 DARCY FLOW

To evaluate diversity, we generate 1000 darcy flow samples for each experiment, aggregate them by channel (permeability $K$ and pressure $p$). and compute a kernel-density estimation. To evaluate sample quality, we generate 10000 samples and evaluate the residual error $\epsilon$. Figure 4 shows the

results of the diversity evaluation, figure 5 the results of the quality evaluation. These highlight several key findings:

1. Continuous direction and angle (as described in section 4.2) lead to increase in sample quality without significant loss of diversity.

2. Discretizing and limiting the number of directions (as described in section 4.3) leads to improved sample quality, but can lead to significant loss in diversity beyond a certain threshold.

3. Angle range and number of discretization bins have little effect on quality or diversity.

4. Incorporating information from all channels by using an n-fold rotation (a double rotation in this case, as described in section 4.4) leads to a measurable increase in sample quality, without loss of diversity.

Furthermore, we investigate the effect of isoclinic rotations. As discussed in section 3.2, isoclinic rotations are a special form of n-fold rotations, where the angles of all planes of rotations are equal to each other. To that end, we extend the action space of experiment 4 (section 4.4) to $(j, k, \Phi_1, \Phi_2)$, where $j, k \in 1, \ldots, d^2$ and $\Phi_i = \frac{k\pi}{79}$, for $k \in 0, \ldots, 79$. Essentially, we add one action dimension for the second angle. The results show that the effect is minimal: the isoclinic rotation converges slightly faster and achieves a slightly higher average residual error ($\sim 1.7\%$). We believe that this stems mostly from the added complexity from the extra free parameter $\Phi_2$.

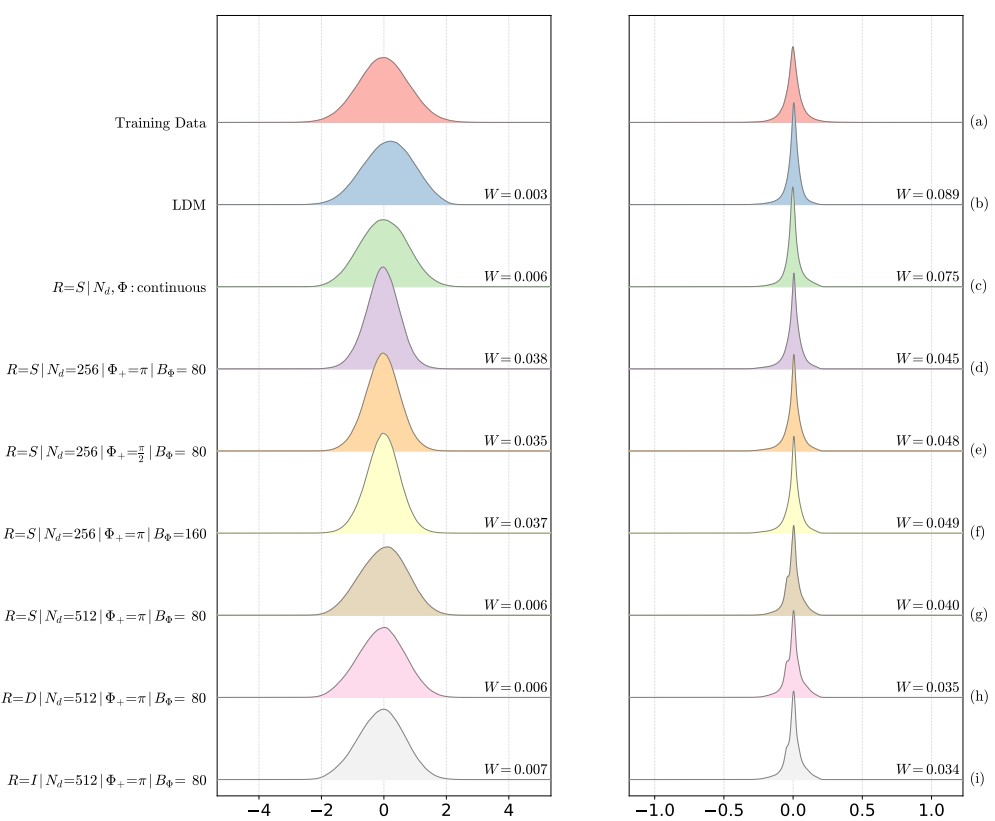

Figure 4: Darcy Flow: Distribution of permeability $K$ (left) and pressure $p$ (right) for different numbers of directions $N_d$, angle ranges $[0, \Phi_+]$, and number of angle discretization bins $B_\Phi$. $R$ represents the type of rotation ($S$: single, $D$: double, $I$: isoclinic) and $W$ denotes the Wasserstein distance to the training data. Plot (c) corresponds to experiment 2, plots (d,e,f,g) correspond to variations of experiment 3, and plots (h,i) correspond to variations of experiment 4.

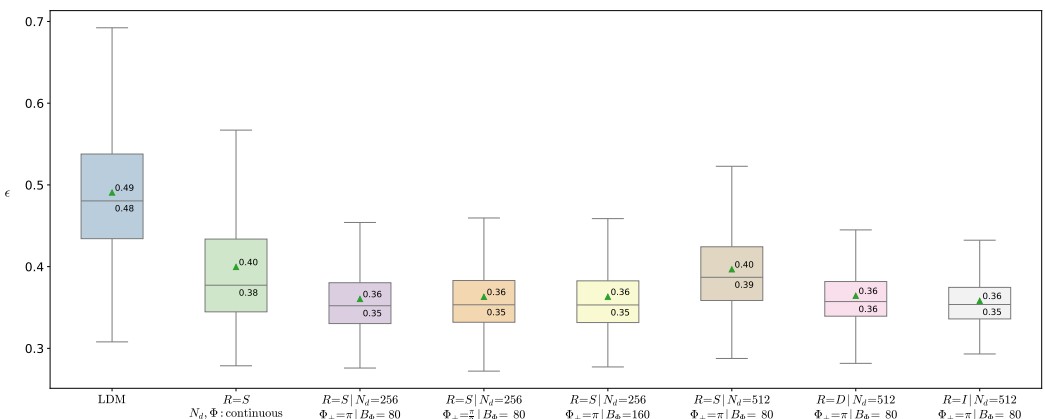

Figure 5: Darcy Flow: residual error for different numbers of directions $N_d$, angle ranges $[0, \Phi_+]$, and number of angle discretization bins $B_\Phi$. $R$ represents the type of rotation ($S$: single, $D$: double, $I$: isoclinic).

## B.2 VORONOI

To evaluate diversity, we generate 1000 images, automatically detect the position of the region centroids and plot their distribution on the $64 \times 64$ pixel grid. To evaluate sample quality, we generate 10000 samples and evaluate the residual error $\epsilon$. Figure 6 shows the results of the diversity evaluation, figure 7 the results of the quality evaluation. These partly mimic the results from section B.1, with a few important differences:

1. Discretizing direction and angle always leads to significant loss in diversity.

2. Both increasing the angle range and the number of discretization bins increase sample quality and diversity.

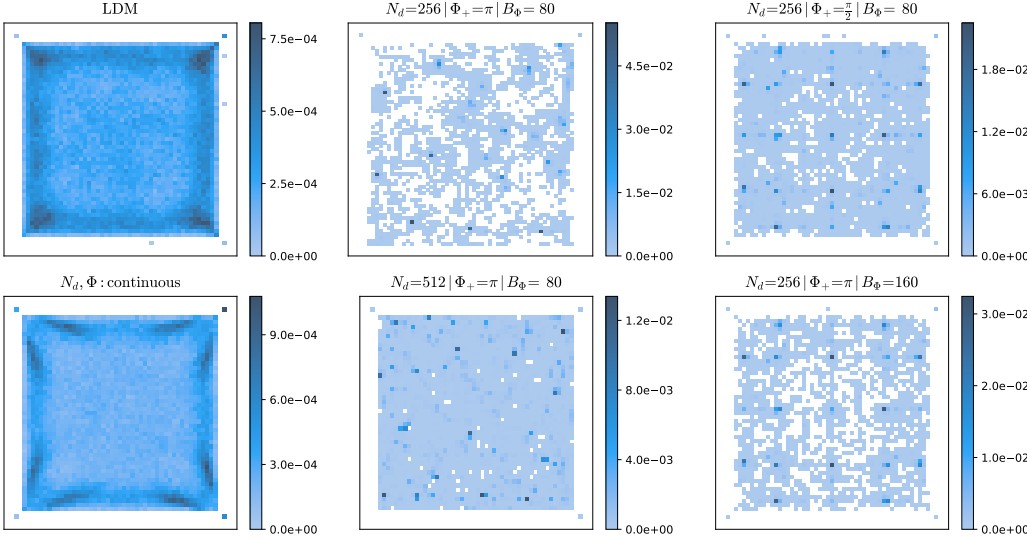

Figure 6: Voronoi: Distribution of region centroids for different number of directions $N_d$, angle ranges $[0, \Phi_+]$, and number of angle discretization bins $B_\Phi$.

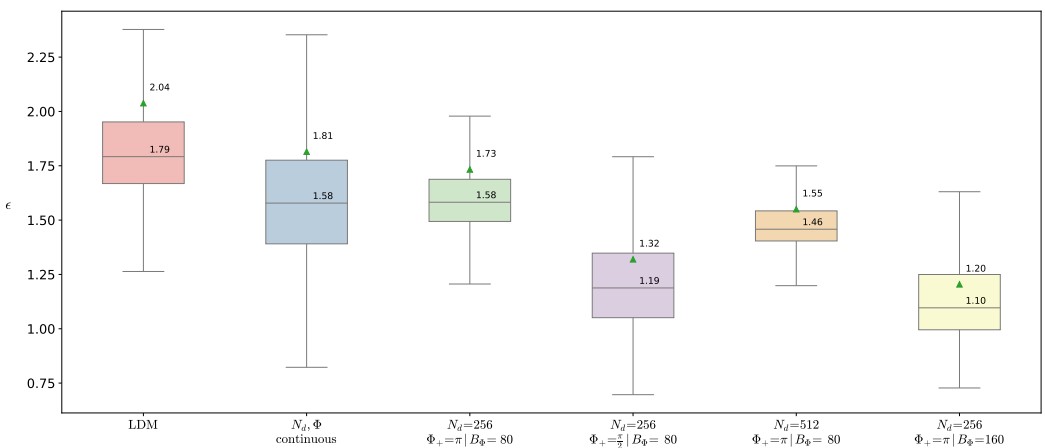

Figure 7: Voronoi: residual error for different numbers of directions $N_d$, angle ranges $[0, \Phi_+]$, and number of angle discretization bins $B_\Phi$.

### B.3 TEXT-CONDITIONED LDM

To evaluate quality, we generate $10000$ images and evaluate the residual error $\epsilon$. Results are shown in figure 3. To evaluate diversity, we generate $1000$ images with RLSO and $10000$ images with the LDM. We encode both generated datasets with CLIP and measure the mean pairwise cosine similarity $\delta$ between encodings within each datasets. We report $\delta = 0.8457$ for the LDM and $\delta = 0.8857$ for RLSO. This indicates a slightly reduced diversity, but no signs of mode collapse, as also confirmed by visual inspection. Furthermore, we investigated how well RLSO trained on a fixed prompt generalizes to unseen prompts and slight architecture modifications. Figures 8 (a,b,c) show that RLSO can generalize to unseen prompts when trained with a fixed one, albeit with a reduced improvement in sample quality. Similarly, it can cope with dropping the Hyper-SD LoRA with which it was trained at inference time, also with a slightly reduced effectiveness.

## C BASELINE COMPARISON: PROJECTION-BASED UPDATES

To further validate the hypothesis that rotations constitute an efficient exploration strategy, we compare RLSO against a popular norm-preserving optimization method used in LSO (similar to e.g. Wallace et al. (2023)). We construct a baseline with an action space analogous to that of Experiment 3 (Section 4.3); however, instead of applying a rotation, this baseline projects the update back onto the hypersphere. The actions lie in the space $(j, \alpha)$, where $j \in 1, \dots, N_d$ and $\alpha = \frac{p}{8}$, for $p \in 0, \dots, 79$. The next state is then computed according to:

$$\mathbf{s}_{t+1} = \sqrt{D} \cdot \frac{\mathbf{s}_t + \hat{\mathbf{h}}_\mathbf{j}\alpha}{\left\|\mathbf{s}_t + \hat{\mathbf{h}}_\mathbf{j}\alpha\right\|_2}$$

where $\hat{\mathbf{h}}_\mathbf{j}$ is th $j$th column of the direction matrix $H \in \mathbb{R}^{d^2 \times N_d}$, $\alpha$ is a step size, and $D = d^2 = 256$ denotes the dimensionality of the vectorized latent space.

Figure 9 demonstrates that RLSO significantly outperforms this norm-constrained baseline. We attribute this performance gap to the limitations of projection-based exploration, which is inherently restricted in its angular reach compared to rotational updates.

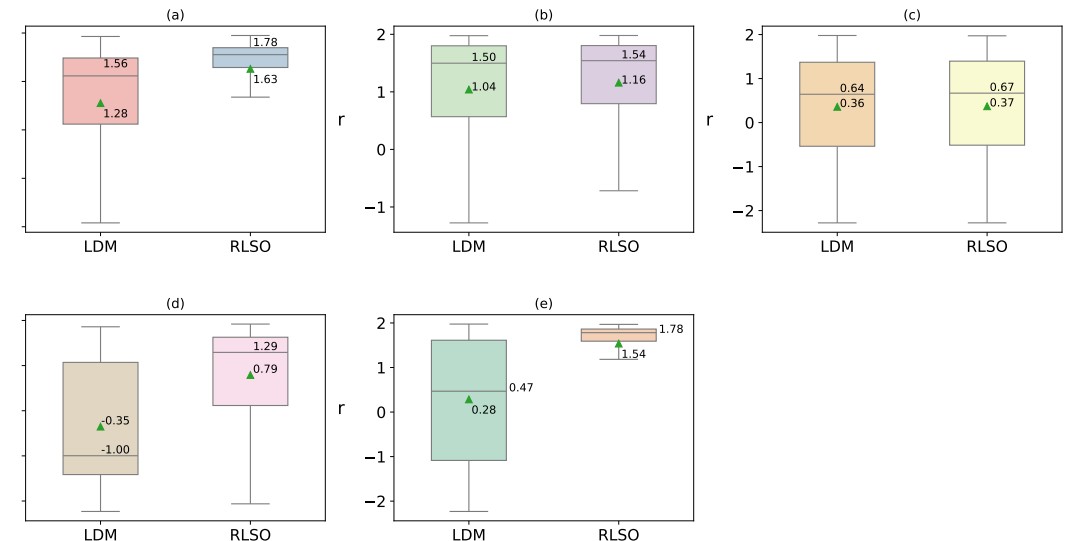

Figure 8: (a) Training with the fixed prompt *"a photo of an astronaut riding a horse on mars."*, reward evaluation with the same fixed prompt. (b) Training with the fixed prompt *"a photo of an astronaut riding a horse on mars."*, reward evaluation with fixed prompt *yellow cat sitting on a park bench.* (c) Training with the fixed prompt *"a photo of an astronaut riding a horse on mars."*, reward evaluation with dynamic prompt, sampled at each inference step from the DrawBench dataset (Saharia et al., 2022). (d) Training with the fixed prompt *"a photo of an astronaut riding a horse on mars."*, reward evaluation with the same fixed prompt, without the Hyper-SD LoRA. (e) Training with the fixed prompt *"a green colored rabbit."*, reward evaluation with the same fixed prompt.

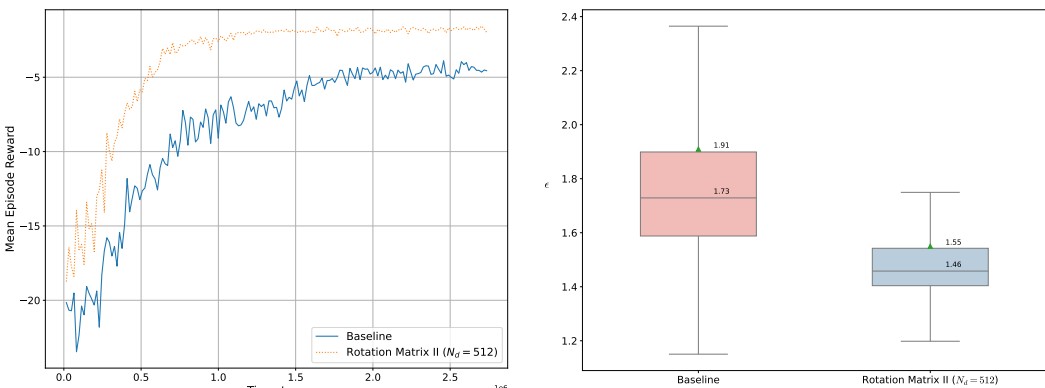

Figure 9: Voronoi: RL training (left) and residual error comparison (right), evaluated on 10000 samples

## D    QUALITATIVE RESULTS

Figures 10 to 14 show a qualitative comparison between the LDM and RLSO. For each comparison, we generated 1000 samples with the LDM and 1000 samples with RLSO and respectively selected 5 at random using a pseudo-random number generator.

## E    LLM USAGE

We used a large language model only for minor editing, such as correcting typos, fixing grammatical errors, and limited rephrasing.

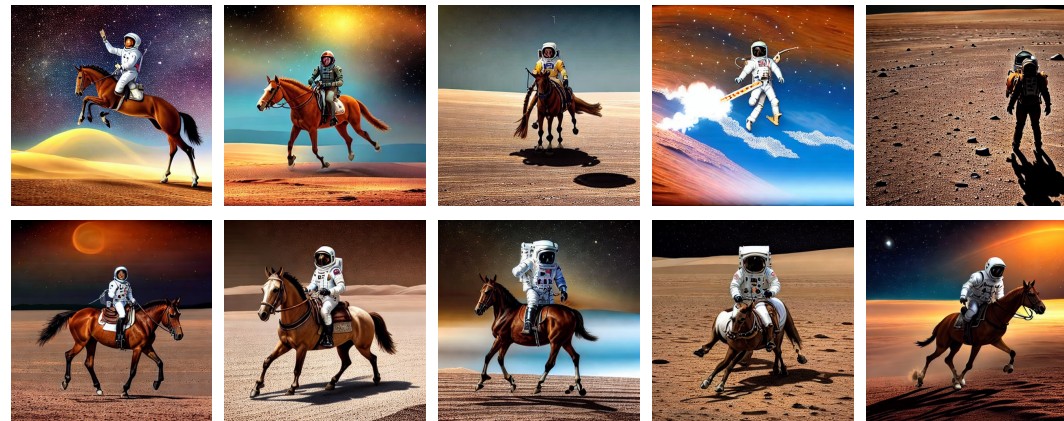

Figure 10: Images generated by the text-conditioned LDM with the prompt *"a photo of an astronaut riding a horse on mars"* (top). The mean reward for the LDM images is $r = 1.09$ (top). Images generated by RLSO (bottom). The mean reward for the RLSO samples is $r = 1.525$.

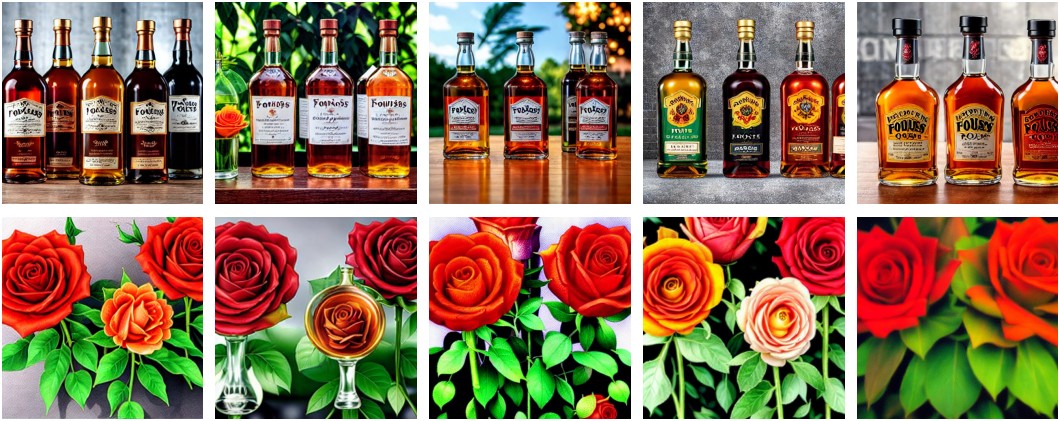

Figure 11: Images generated by the text-conditioned LDM with the prompt *"four roses."* (top). The mean reward for the LDM images is $r = -1.08$ (top). Images generated by RLSO (bottom). The mean reward for the RLSO samples is $r = 1.217$. Note that RLSO avoids the SD 1.5 failure mode associated with the whisky brand of the same name.

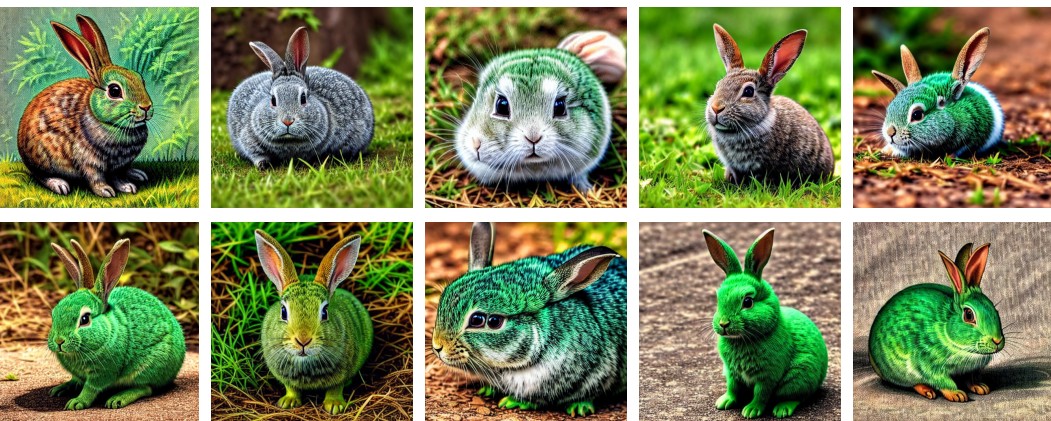

Figure 12: Images generated by the text-conditioned LDM with the prompt *"a green colored rabbit."* (top). The mean reward for the LDM images is $r = -0.204$ (top). Images generated by RLSO (bottom). The mean reward for the RLSO samples is $r = 1.733$.

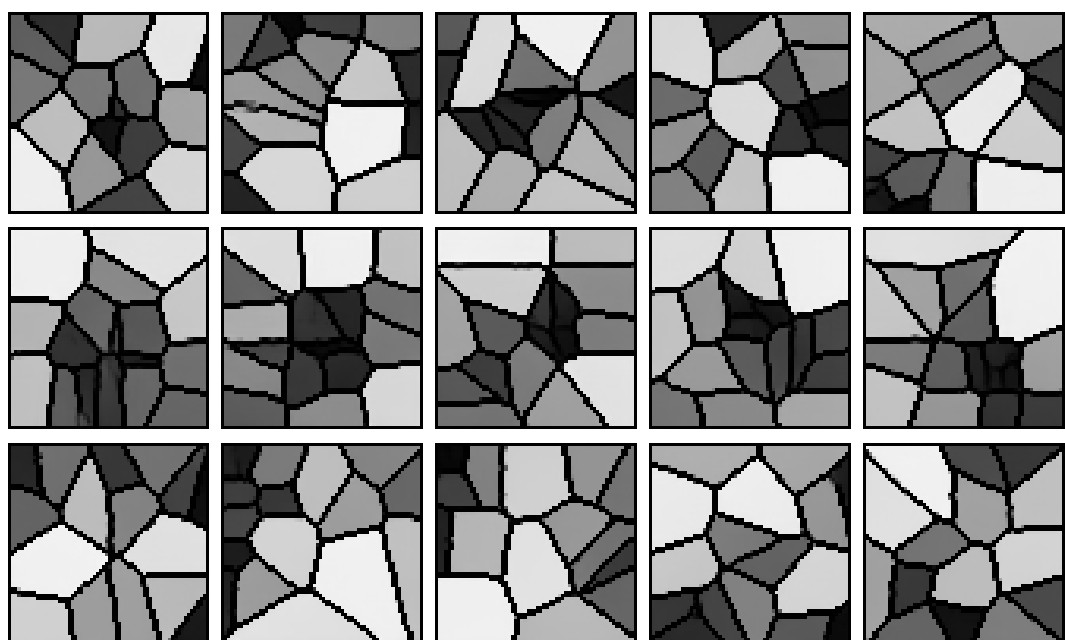

Figure 13: Voronoi samples generated by the unconditional LDM. The mean residual for the LDM samples is $\epsilon = 1.879$ (top). Voronoi samples generated by the baseline for Experiment 1. The mean residual is $\epsilon = 1.864$ (middle). Voronoi samples generated by RLSO for Experiment 2 (continuous action space). The mean residual for the RLSO samples is $\epsilon = 1.51$ (bottom).

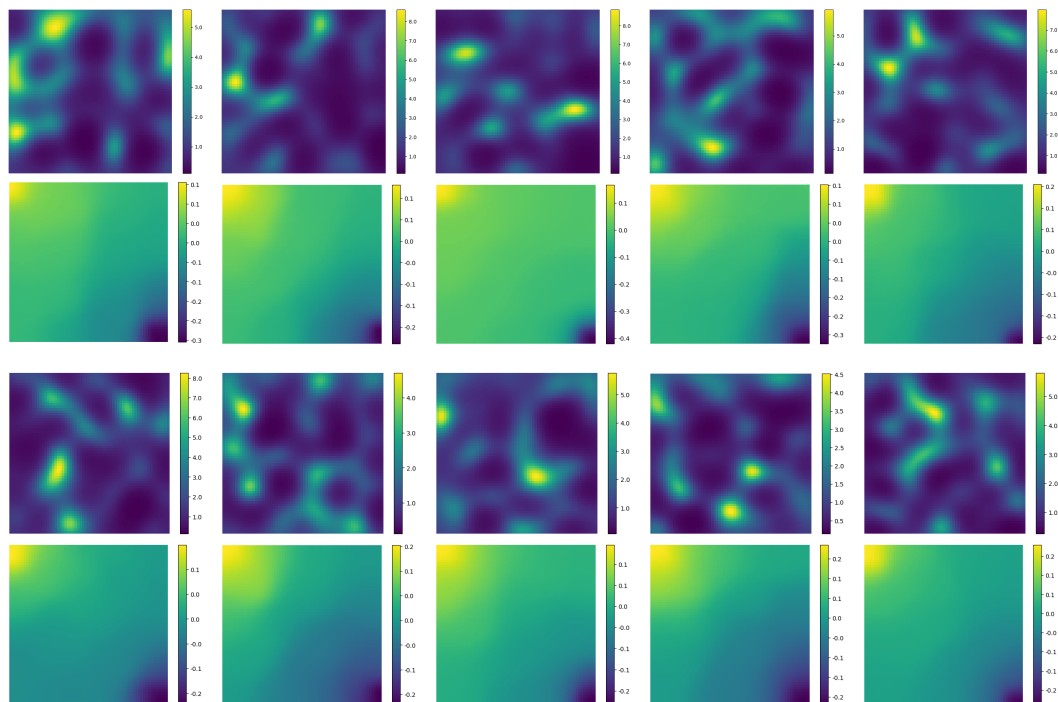

Figure 14: Darcy Flow permeability fields $K$ and pressure fields $p$ generated by the unconditional LDM (top two rows). Row 1 shows samples of $K$, and Row 2 shows samples of $p$. The mean residual for the LDM samples is $\epsilon = 0.552$. Darcy Flow permeability and pressure fields generated by RLSO for Experiment 3 ($N_d = 512$) are shown in the bottom two rows. Row 3 shows $K$, and Row 4 shows $p$. The mean residual for the RLSO samples is $\epsilon = 0.424$.

