# OpenReview forum: "Rotations on Latent Hyperspheres: a Geometry-Aware Guiding Framework for Diffusion Models"
_ICLR.cc/2026/Conference — Submitted to ICLR 2026_

### Official Review · Reviewer_5bpF · 2025-10-28

**Soundness:** 3
**Presentation:** 3
**Contribution:** 3
**Rating:** 6
**Confidence:** 3

**Summary:**

This paper introduces Rotational Latent Space Optimization (RLSO), a framework for effective latent space exploration. The core idea is to replace additive perturbations in the latent space with geometry-aware rotations, leveraging the inherent spherical structure of high-dimensional Gaussian priors. This exploration is amortized using a Reinforcement Learning agent, enabling efficient, single-pass inference. The method is validated on two distinct domains—solving a physics-based PDE (Darcy Flow) and generating structured images (Voronoi)—demonstrating improved sample quality according to domain-specific reward functions.

**Strengths:**

1. Compelling Motivation for AI-for-Science.
2. Principled Methodological Core. The key insight that high-dimensional Gaussian latents concentrate on a thin spherical shell and should be explored via rotations rather than additive steps is mathematically sound.
3. The use of two different domains effectively demonstrates the generality and versatility of the proposed RLSO framework.

**Weaknesses:**

1. Oversimplification of Prior LSO Work. The paper's motivation partly relies on the claim that additive perturbations "destroy the statistical structure" of the latent space by moving samples away from the shell. In practice, many optimization-based LSO methods use small step sizes, gradient information, or explicit regularization (e.g., projecting back to the shell) to remain in high-probability regions.
2. Lack of qualitative results for the unconstrained exploration baseline (Experiment 1)
3. Lack of comparison to other SOTA LSO methods

**Questions:**

Please address the concerns raised in the weakness.
Please elaborate on how the hyperparameters and the weighting scheme were determined in residual error?
Please provide a physical intuition for why applying isoclinic rotation leads to more effective exploration?
Further, the paper focuses on exploring the prior distribution. However, we believe the real goal is to explore the data manifold. A key discussion point is: does rotational exploration merely yield latents with a high prior probability, or does it more effectively navigate the data manifold? The authors should discuss this distinction. Does the rotation operation have a meaningful interpretation as moving along the data manifold, compared to an additive step?

---

### Official Review · Reviewer_nXLK · 2025-10-31

**Soundness:** 2
**Presentation:** 2
**Contribution:** 2
**Rating:** 2
**Confidence:** 3

**Summary:**

The authors consider the problem of optimizing over initial conditions (noise realizations) of a diffusion model to generate samples that satisfy certain constraints, encoded in terms of a reward / constraint function quantifying the residual error.  In particular, they propose to utilize reinforcement learning (PPO) to optimize over rotations in latent space.  The main idea is to use a parametrization over n/2 mutually orthogonal planes.  The approach is demonstrated on two synthetic data sets – solutions of a Darcy Flow PDE, and a collection of synthetic images, where optimizing over the rotations outperforms unconstrained exploration, and the particular chosen parametrization of the rotations offers advantages.

**Strengths:**

+ efficient, constrained generation with diffusion models is an important problem

**Weaknesses:**

- Fine-tuning and guidance of diffusion models (also via RL) is extensively studied, and the authors do not compare against any prior methods (such as classifier-free guidance etc.)
- The considered experimental settings seem rather simplistic, and there is no evidence that the approach would scale to more complex domains.
- Given the primarily empirical nature of the paper, I don't find the empirical evaluation particularly convincing.

**Questions:**

- How were the reward thresholds set? (Section 3)
- Why is there no comparison with established methods (perhaps suitably adapted) for tuning diffusion models?

---

### Official Review · Reviewer_8AZh · 2025-11-01

**Soundness:** 3
**Presentation:** 2
**Contribution:** 3
**Rating:** 6
**Confidence:** 5

**Summary:**

To address the limitation that diffusion models—despite their power across domains—often generate samples deviating from domain-specific constraints due to their data-driven nature, this paper proposes a plug-and-play Reinforcement Learning (RL)-based framework called Rotational Latent Space Optimization (RLSO). Operating in the latent space of pre-trained frozen diffusion models, RLSO leverages the near-spherical geometry of high-dimensional Gaussian distributions and employs a novel rotation-matrix-based scheme for efficient latent space exploration. Guided by task-specific rewards computed on final samples, the framework steers the model toward feature-preserving outputs.
Experiments are conducted on two diffusion models: one trained on solutions of the Darcy Flow PDE and another on a synthetic Voronoi dataset with complex structural features. Results demonstrate significant improvements in sample quality: a ~25% relative reduction in PDE residual for the Darcy Flow task and up to a ~44% relative improvement in the feature-alignment metric for the Voronoi dataset, compared to vanilla diffusion models. Additionally, rotation-matrix-based exploration outperforms unconstrained exploration, validating the effectiveness of the geometry-aware approach for latent space control.

**Strengths:**

1. The paper innovatively integrates high-dimensional rotation matrices with RL for latent space guidance of diffusion models. Unlike existing methods that treat the latent space as a generic Euclidean space, it explicitly leverages the near-spherical geometry of Gaussian latent spaces, filling a critical gap in current Latent Space Optimization (LSO) techniques.
2. Adopting a plug-and-play architecture, RLSO avoids fine-tuning pre-trained diffusion models. By amortizing optimization costs via RL, it generates constraint-aligned samples with only a single denoising pass at inference time, solving the high computational overhead issue of iterative LSO methods.

**Weaknesses:**

1. Trade-off Between Quality and Diversity: While sample quality is improved, the paper observes a narrowing of the sample distribution and an increased risk of mode collapse (more pronounced in the Voronoi dataset). No effective mechanism is proposed to balance sample quality against diversity.
2. Scalability to Higher Dimensions: Although the modular design partially mitigates high-dimensional challenges, the training efficiency and exploration effectiveness of the RL agent are not fully validated as the latent space dimension increases with model scale.
3. Limited Exploration of Rotation Types: The study focuses only on simple rotations and isoclinic double rotations, without systematically investigating how other rotation types (e.g., non-isoclinic double rotations) impact optimization performance. This limits the generalization of the proposed method.
4. Narrow Application Scope: Experiments are restricted to PDE solution generation and synthetic image generation. The framework is not validated in constraint-intensive complex scenarios (e.g., molecular generation, engineering design), leaving its practical applicability unproven in broader fields.

**Questions:**

1.Regarding the trade-off between sample quality and diversity, are there specific parameter adjustment strategies (e.g., optimizing the range of rotation angles or the number of search directions) that can improve diversity without significantly compromising quality?
2. As the latent space dimension increases substantially (e.g., several times larger than the current 16×16), what trends will emerge in the training time and computational resource consumption of the RLSO framework?
3. For scenarios with different data distribution characteristics, is there a clear criterion for selecting rotation types (e.g., simple rotation vs. double rotation)?
4. When the framework is applied to multi-objective optimization scenarios (e.g., satisfying both physical constraints and style/text prompts simultaneously), do the reward function design and rotation strategy require adjustments? If so, what specific adjustments are needed?
5. What are the quantitative differences in computational complexity and convergence speed between RLSO and mainstream constraint-guided diffusion model methods (e.g., score matching, energy model guidance) in the current field?

---

### Official Review · Reviewer_ENjc · 2025-11-01

**Soundness:** 2
**Presentation:** 1
**Contribution:** 1
**Rating:** 2
**Confidence:** 3

**Summary:**

The paper proposes Rotational Latent Space Optimization (RLSO), a geometry-aware guidance framework that treats optimization of a diffusion model’s initial noise as an RL problem and explores the Gaussian latent on its hypersphere via rotation matrices, enabling constraint-aligned samples with a single denoising pass.
Using Mortari’s n-D rotation construction, the policy rotates latent codes along geodesics, preserving norm and better respecting in-distribution structure than unconstrained Euclidean perturbations.
On Darcy Flow PDE solutions and a Voronoi synthetic dataset, RLSO yields substantial gains.

**Strengths:**

1. Geometry-aware control: the proposed approach respects the hyperspherical geometry of Gaussian latents via norm-preserving rotation matrices, giving a principled, modular way to steer along geodesics and decouple “direction vs. angle” in exploration.

2. Efficiency at inference: the proposed approach produces samples in a single denoising pass.

3. Empirical gains: On Darcy Flow it achieves ~25% relative reduction in PDE residual vs. vanilla LDM, and on Voronoi up to ~44% improvement on the feature metric; restricting rotations to fixed Hadamard directions converges faster and often higher.

**Weaknesses:**

The presentation of the paper is not clear. After reading the paper several times, I still cannot understand procedure of the proposed algorithm. I hence cannot recommend acceptance, even though the empirical results seem to suggest the effectiveness of the algorithm.

I will now describe my current understanding of the problem formulation, the idea, and the role of RL in the algorithm.

1. The goal of the project is to improve the diffusion model so that the generated samples satisfy `domain-governing constraints`.
2. Instead of enforcing the `domain-governing constraints` at the terminal of the inference time, this papers considers the LSO type method, which adjusts the initial random noise of the inference procedure. [This is already confusing for me: Do you mean to change the distribution of the initial random noise? If so, it seems to contradict with the basic principle of diffusion model.]
3. The authors observe that since the initial random noise are Gaussian distributed and hence due to the concentration of measure phenomenon, the initial random noise should concentrate on the sphere with a proper radius.
4. Given the observation in point 3, the authors propose to formulate the LSO problem as an RL problem where the action space consists of rotations so that the state space remains the said sphere after performing actions. [I cannot understand how the `domain-governing constraints` (supposedly only meaningful at the terminal of the inference phase), are translated to the reward of the action on the initial random noise. A simple strategy would be to rollout the inference trajectory every time one needs to evaluate the reward. However, this seems to be a very expensive way of training. Also, this simple strategy is clear not novel.]

**Questions:**

Please see the discussion above.

---

### Meta-Review · Area_Chair_mnHS · 2026-01-05

**Summary:**

The paper proposes a geometry-aware framework for guiding diffusion models by optimizing their initial latent noise using RL. While the rebuttal addresses many concerns, doing so would require substantial revisions to the main paper, particularly to better integrate clarifications and expanded experimental analyses. Therefore, I encourage the authors to resubmit a revised version with improved clarity and a stronger evaluation and positioning of the method.

**Reviewer Concerns:**

Reviews ENjc and nXLK pointed out the clarify issue and lack of experimental comparisons in various settings including a large scale, complex domain. Authors efforts address some of issues raised by reviews. Thus, presented experimental evaluations are not enough.

**Reviewer Scores:**

Reviews ENjc and nXLK may improve the scores. However, the margin would be limited.

---

### Decision · Program_Chairs · 2026-01-26

Reject